# Gender Matters: A Gender Analysis of Healthcare Workers' Experiences during the First COVID-19 Pandemic Peak in England

**Nina Regenold** [1] and **Cecilia Vindrola-Padros** [2,*]

1 Department of Anthropology, University College London, 14 Taviton Street, London WC1H 0BW, UK; nina.regenold.19@ucl.ac.uk
2 Department of Targeted Intervention, University College London, Charles Bell House 43-45 Foley Street, London W1W 7TY, UK
* Correspondence: c.vindrola@ucl.ac.uk

**Abstract:** The coronavirus (COVID-19) arrived in the United Kingdom (UK) in February 2020, placing an unprecedented burden on the National Health Service (NHS). Literature from past epidemics and the COVID-19 pandemic underscores the importance of using a gender lens when considering policy, experiences, and impacts of the disease. Researchers are increasingly examining the experiences of healthcare workers (HCWs), yet there is a dearth of research considering how gender shapes HCWs' personal experiences. As the majority of HCWs in the UK and worldwide are women, research that investigates gender and focuses on women's experiences is urgently needed. We conducted an analysis of 41 qualitative interviews with HCWs in the British NHS during the first peak of the COVID-19 pandemic in the Spring of 2020. Our findings demonstrate that gender is significant when understanding the experiences of HCWs during COVID-19 as it illuminates ingrained inequalities and asymmetrical power relations, gendered organizational structures and norms, and individual gendered bodies that interact to shape experiences of healthcare workers. These findings point to important steps to improve gender equality, the wellbeing of healthcare workers, and the overall strength of the NHS.

**Keywords:** gender; healthcare workers; gender equality; United Kingdom; COVID-19; National Health Service

## 1. Introduction

Since it began to spread in Wuhan, China at the end of 2019, Corona Virus Disease 2019 (COVID-19) has tested the strength of health systems worldwide. Healthcare workers (HCWs) have worked tirelessly on the frontlines, risking their lives to protect people from this unprecedented threat. The virus has forced nations to reckon with deep-rooted social inequalities that have led to differential impacts, risks, and deaths from COVID-19.

Literature from past epidemics and the current COVID-19 pandemic points to the importance of using a gender lens when examining policy, experiences, and impacts of the disease. Gender analyses from the Ebola outbreak point to the concentration of women in informal and formal care roles, the resulting disproportionate impact of the disease on women, the need for the appreciation of women and their contribution to fighting the epidemic (Johnson and Vindrola-Padros 2014), the lack of representation of women in global health leadership, and the importance of addressing gender differences and gendered impacts of epidemics (Harman 2016). Nevertheless, less than 1% of the combined research published on Zika and Ebola considers gender (Smith 2019).

There has been a substantial call for attention to gender during the COVID-19 pandemic, reflecting a concern that responses to the pandemic that fail to consider gender differences and norms will be ineffective and uphold current gender and health inequalities

(Wenham et al. 2020). Researchers attending to gender have so far established that men are more likely to have severe outcomes from COVID-19 than women (Rabin 2020; Purdie et al. 2020), women often occupy positions of high risk due to their overrepresentation in the healthcare sector, and there is an inadequate representation of women in health leadership and decision-making roles across the globe (Kim et al. 2020; EIGE 2020). Being female and a nurse have both emerged as associated with a higher prevalence of depression and anxiety compared to other healthcare workers (Pappa et al. 2020). Higher rates of posttraumatic stress symptoms were reported by female HCWs in China (Liu et al. 2020) and certain groups of female HCWs in Italy (Di Tella et al. 2020) during the COVID-19 pandemic. Lastly, in many countries, women HCWs account for a higher proportion of COVID-19 infections compared to men (Miyamoto 2020).

Globally, females account for 70% of the health and social care sector, and a large majority of nurses and midwives (Boniol et al. 2019). In the UK National Health Service (NHS), 77% of total staff are female (NHS Digital 2018a). Despite a growing body of research on HCWs during COVID-19, there is a dearth of research considering how gender shapes HCWs' experiences. As women account for the majority of HCWs both worldwide and in the UK, research that investigates gender and, particularly, women's experiences is urgently needed. This paper examines the experiences of HCWs during the first peak of the COVID-19 pandemic in England through a gender lens, incorporating literature from the social sciences and feminism, health systems research, past epidemics, and the current COVID-19 pandemic to discuss our findings and outline future changes.

### 1.1. Our Approach to Gender

Our approach to gender is based on relational theory, which conceptualizes gender as multidimensional, "embracing at the same time economic relations, power relations, affective relations and symbolic relations; and operating simultaneously at intrapersonal, interpersonal, institutional and society-wide levels" (Connell 2009; Lorber 1994 cited in Connell 2012). Gender is also intersectional, as it interacts with other forms of social difference such as race, class, ethnicity, age, etc. (Springer et al. 2012, p. 1661).

Incorporating a gender lens allows us to understand the ways in which gender power relations shape different levels of experience, from individual gendered bodies to macro-level gendered norms and inequalities. As noted by Gerson (2004), a gender lens provides a framework to contextualize individual experiences within the larger sociocultural context and structural inequalities. Within this framework, gender can be understood as a social structure that organizes the experiences and opportunities of individuals (Risman 2004; Gerson 2004). The reflexive relationship between individual action and larger structures is crucial in understanding how gender inequalities are reproduced, resisted, or perpetuated (Scarborough and Risman 2017).

### 1.2. Gender and Healthcare Workers

Within health systems research, gender is understood as a critical social stratifier that influences the positioning of women and men within healthcare structures and their experiences within that location (Morgan et al. 2016; George 2007). Regardless of the gender of individuals who occupy the medical positions, gender silently structures power, vocational identity, and the division of labour among medical professionals (Hinze 1999; Risberg 2004).

Literature on gender analysis of health systems and human resources for health has revealed gender biases and inequalities in occupational biases, descriptions of health work, substitution and delegation, and inequalities in pay among HCWs that disadvantage women (George 2007). Research using qualitative designs has revealed gender differences in patterns of employment, access to training, and coping strategies among HCWs in fragile and post-conflict contexts (Witter et al. 2017), gender inequalities and discrimination within the health workforce in Rwanda and Kenya (Newman et al. 2011a, 2011b), a gendered culture of medicine that shapes the careers and experiences of hospital consultants in the

UK (Jefferson et al. 2015; Dumelow et al. 2000), and how gender is inscribed on the body and thus structures working relations among doctors and nurses in Sweden (Davies 2003).

### 1.3. Gender and the National Health Service

Despite efforts towards equality, the NHS remains stratified along gender lines. The tendency for men to occupy positions of greater income, prestige, and power than women is referred to as vertical segregation (Risberg 2004). Women account for 77% of the NHS workforce but only 47% of 'very senior manager' roles (NHS Employers 2019). This gap is significantly wider for ethnic minority women: as of 2017, Asian/British Asian women accounted for 1% and Black/Black British Women accounted for 0.5% of all very senior manager roles in the NHS (NHS Digital 2018b). This vertical segregation contributes to the NHS gender pay gap of 23% (Department of Health and Social Care 2019).

In the NHS, women make up the large majority (89%) of all nurses and health visitors (NHS Digital 2018a). Black, Asian and Minority Ethnic (BAME) individuals account for 20% of NHS nurses and support staff (Cook et al. 2020) and the majority of NHS nurses in London (RCN 2018). Internally, nursing is socially stratified along ethnic, gender, and class lines (Batnitzky and McDowell 2011; George 2007).

Lastly, numerous cases of sexism and bullying of female doctors have been reported in the NHS (BMA 2019; Triggle 2019).

## 2. Methods

The interview data used for this gender analysis was gathered as part of an ongoing project by the Rapid Research, Evaluation and Appraisal Lab (RREAL) at University College London (UCL). The study is designed as a rapid appraisal and combines media and social media analysis, policy review, and telephone interviews to assess HCWs' experiences and perceptions during the COVID-19 pandemic (Vindrola-Padros et al. 2020). A purposive sample of HCWs was selected to take part in the RREAL study based on a sampling framework that included a variety of HCWs of different specialties and ranks. Participants were selected from multiple sites in England, including two acute care trusts, one community care trust, and a specialist trust. All participation in this study was voluntary. Participants were sent information sheets and written informed consent was given.

Interviews were conducted over the telephone and recorded with the consent of participants. The interviews were semi-structured and investigated HCWs' general experiences and perceptions of COVID-19, including questions about preparedness strategies, mental health support, impact of COVID-19 on health services, experiences with personal protective equipment (PPE), and home life (see Appendix A for the interview topic guide).

The sample used for this specific analysis consists of 41 interviews carried out between 21 May and 19 July 2020. Interviewees reflected on their recent experiences during the peak of COVID-19 in the Spring of 2020 as well as their current experiences caring for COVID-19 patients. The sample was roughly one quarter male and three quarters female—a proportion similar to that of NHS England. Interviewees were all involved in health services during COVID-19 and occupied a variety of roles and ranks, including nurses (specialist, research, lead) anaesthetists, surgeons, allied health professionals (AHPs) (physiotherapists, dieticians, occupational therapists), and both trainee and consultant doctors. See Table 1 for a full list of the sample characteristics.

**Table 1.** Sample Characteristics (*n* = 41).

| | |
|---|---|
| **Gender** | |
| Male | 9 |
| Female | 32 |
| **Age** | |
| 20s | 8 |
| 30s | 21 |
| 40s | 6 |
| 50s | 6 |
| **Race/Ethnicity** | |
| White British/Irish | 27 |
| White Other | 5 |
| Asian/British Asian | 5 |
| Black British | 1 |
| Mixed Race | 4 |
| White and Black African | 2 |
| White and Asian | 2 |
| **Role** | |
| Doctor | 17 |
| Allied Health Professionals | 13 |
| Nurse | 10 |
| Pharmacist | 1 |
| **Caring Responsibilities** | |
| Child(ren) | 15 |
| Elderly Parent | 1 |
| Pregnant (or recently pregnant) | 4 |
| **Location** | |
| London Trust 1 | 14 |
| London Trust 2 | 21 |
| Other NHS Trust | 2 |
| Private Hospital | 1 |
| **Education Level** | |
| Undergraduate Degree | 13 |
| Postgraduate Degree | 26 |
| Unknown | 2 |
| **Years in Service** | |
| $\leq 5$ | 5 |
| 6–10 | 13 |
| 11–19 | 12 |
| 20+ | 8 |
| Unknown | 3 |

Four healthcare workers in this sample were pregnant or recently pregnant. These interviews were less structured and investigated HCWs' experiences being pregnant during the pandemic. Areas covered included relevant policies or guidelines, how the HCW's safety was managed, emotions at work and home, and support provided.

*Data Extraction and Analysis*

One researcher (NR) performed selective transcriptions (verbatim) of twenty interviews. As interviews covered a broad range of topics, specific details about health services and COVID-19 patients that were not related to personal experience or power relations were excluded. Twenty-one interviews were transcribed by an external service, and irrele-

vant information was eliminated from these in the same manner. Anonymity was sustained throughout analysis and in the presentation of findings.

We used a combined inductive and deductive approach to thematic analysis, meaning that codes and themes emerged both from the raw data and from pre-existing literature (see coding framework in Appendix B). Morgan et al.'s (2016) framework for gender analysis in health systems research was used to locate areas relevant to gender power relations. This framework asks the researcher to interrogate the following domains: "*Who has what* (access to resources); *who does what* (the division of labour and everyday practices); *how values are defined* (social norms, ideologies, beliefs and perceptions), and *who decides* (rules and decision-making)" (Morgan et al. 2016, p. 5).

The software NVivo version 12 (QSR International) was used to conduct thematic analysis of the interview data. Codes arising from the first 5 interviews were used to develop an analytical framework for the rest of the data based on relevance to gender and salience. One researcher (NR) coded the rest of the data, adding additional codes that arose as important and relevant. Another researcher (CVP) cross-checked the coding framework. Framework analysis was also used through the node matrix feature on NVivo to understand data in relation to the characteristics and positionality of each respondent, and to draw connections between respondents and themes (Gale et al. 2013).

At the time of the interviews, the peak had largely passed but many HCWs were still in their redeployed positions and caring for COVID-19 patients. For the purpose of this paper, 'peak' refers to the weeks during the first wave of COVID-19 in which cases were highest in England and hospitals were overwhelmed with patients, during the end of March, April, and May (Appleby 2020). Interviewees largely reflected on their experiences during the first peak of the COVID-19 outbreak in the UK in April, and so the following findings apply mostly to that time period.

## 3. Results

### 3.1. Division of Labour

There was a consistent difference between the work experiences of lower-ranking and higher-ranking healthcare workers. Management and higher-ranking staff were generally less involved in clinical work than junior staff, meaning they spent less time in PPE, likely had a lower risk of exposure to the virus, and were farther removed from the emotional toll of caring for COVID-19 patients. As a lead nurse explained, "*Because I'm a Line Manager and I wasn't involved in that much clinical work, I think my staff have suffered a lot more than I have [ . . . ] We've got other issues to deal with but the physical bit of being under all this equipment and it's hot and you can't eat and you can't drink . . . It must have been really, really stressful*".

### 3.2. Redeployment

Redeployed staff (those who were transferred to a different facility or role) seemed especially impacted by the hardships that all HCWs faced during the pandemic. The difficulties of working with COVID-19 patients, such as working in PPE and caring for a high volume of severely ill patients, were compounded by inadequate training and preparation and new environments and colleagues. Most (but not all) redeployed staff were given training, but this was brief (e.g., 1 week) and many had to learn on the job under the supervision of more senior staff who were already over-burdened. Redeployed staff often had to work with completely new colleagues, which is consequential as staff reported relying on their team members as crucial sources of support.

Healthcare workers who were not redeployed also faced difficulties, such as the reorganization of services because of COVID-19 and picking up extra work from colleagues who were redeployed. Still, HCWs who were not redeployed felt "*extremely lucky*," "*needed*," and acknowledged that they were in relatively better positions. One consultant surgeon explained this, "*I think that having not been redeployed and being able to do our work . . . it was a most rewarding thing. We were all very much happy to be redeployed and help in whatever areas we*

*were needed, but ... we would have been taken away from what we do the most which is cancer care ... Which is something that definitely would have impacted psychologically to us as well*".

Based on reports from HCWs, those in higher-ranking roles seemed more likely to remain in their original roles. For instance, in the surgical departments, some theatre nurses, ward nurses, and advanced care practitioners were redeployed to ICU while surgeons, and in one case junior surgical staff, remained. Among physiotherapists (PTs), junior staff were mainly redeployed. One consultant anaesthetist in the sample was moved to acute care, but she had volunteered to be relocated. This pattern, however, did not hold for nurses. Nurses were the only group in this sample in which senior staff who did not work in ICU were redeployed to ICU in lower-ranking positions.

### 3.3. Burden on Nurses

Nurses were perceived—both by the nurses themselves and other HCWs—as having a particularly difficult time during the peak of the COVID-19 pandemic. They were often mentioned as an especially stretched, stressed, and over-burdened group. Nurses worked long hours (e.g., 12-h shifts) in PPE, often with reduced breaks during shifts, and for weeks at a time (e.g., working every day for 2 weeks). Nursing staff described unsafe ratios of 1 nurse to 4 or 6 severely ill patients. Due to the 'stretching' of nurses and the limitations of PPE, nurses were unable to provide the normal standard of care that they were proud of. The new approach with patients of "*just touch and go*" was difficult for nurses and resulted in stress and frustration over a lack of control.

Redeployed nurses especially were described as "*the hardest hit*". Nurses were redeployed from a wide variety of backgrounds, meaning that many had little experience in acute care and had to adapt to a sudden and dramatic change in role. An ICU nurse stated, "*all of these nurses who have never been to ITU before were terrified, you could see it on their faces*". Most nurses received training, but this was done quickly, leaving them to learn on the job under the supervision of over-burdened ICU nurses who were also caring for severely ill patients. This created a highly stressful environment for all involved: "*The nurses working on the unit who were quite stressed ... they were the most trained people in their bay but they were working with lots of people who weren't trained in ICU and they were very much having to carry everybody else*".

While challenged physically, nurses placed greater emphasis on the emotional toll of working with COVID-19 patients during the peak. Nurses discussed not being able to "*switch off*" from work when they went home, being anxious about their next shifts, having trouble sleeping, and waking up thinking about work. Crying at work was most commonly reported by nurses; one nurse described a particularly hard night, "*I think everyone that night cried literally every hour on a corner*". Nurses in charge also seemed to carry a heavy emotional burden, even though they were less involved in caring for patients. Nurse leads described supporting and carrying their team, which included "*sucking up a lot of sadness for the team*".

In suggesting changes to the health system, HCWs brought up a need for more support (e.g., education bursary) and higher salaries for nurses.

### 3.4. PPE

The HCWs included in this study came from well-supplied trusts that generally had adequate access to PPE; however, some HCWs reported having inadequate protection. These groups were some AHPs, healthcare assistants, and cleaning staff. HCWs also acknowledged that, although they might have PPE, this was not the case everywhere and friends in other trusts, especially nurses, had difficulties accessing PPE. One nurse stated, "*I've heard several times ... 'don't use this now, we're lucky we have PPE, all the trusts and all the hospitals don't have it' and then you just feel guilty because you know that the nurses aren't protected somewhere else*".

Additionally, a female doctor mentioned male bias in PPE: "*The scrubs, there weren't enough small ones, which I feel is always an example of everyday sexism, you know, for small women*

*like myself . . . There would be just no small ones at all, or the extra, extra-large in the machines, and you think well you wouldn't expect a six-foot man to wear some that would fit me*". A female PT also mentioned that the gowns were very long and a tripping hazard.

### 3.5. Mental Health Support

Staff noted the greater attention to and support for mental health and wellbeing during the COVID-19 pandemic. Although mental health support for staff was provided at most facilities, not all staff were able to benefit from this when they needed it. At one trust, support for staff was not well-organized towards the beginning of the peak, when staff may have needed it the most. Even when support was available, many HCWs had to take time out of their busy work schedules to access this support, which was nearly impossible for over-burdened HCWs. A consultant anaesthetist expressed her concern about the impact of this on certain groups, "*I was quite surprised that there wasn't [psychological support] particularly for the high-risk groups, so the ITU nurses and all the ward staff who are being pushed into ITU, stretching their skills in a very upsetting environment, weren't being given allocated time, even on a fortnightly basis, paid to be there to get this psychological input and support*".

HCWs seemed to rely on colleagues for support and reported feelings of pulling together and camaraderie. One Speech and Language Therapist stated, "*You start to build a bit of a support structure . . . with your colleagues, with the doctors, with the nurses, and you feel that you're kind of one big family in this together and that was quite encouraging*".

### 3.6. Leadership and Decision-Making

Those in positions of power were responsible for preparedness strategies, access to PPE, training, and wellbeing support for staff, all of which shaped HCWs' experiences. Satisfaction with leadership differed among HCWs, but there was consensus over the qualities that constitute strong leadership: early preparation, transparency, a sense of reciprocity, trust, open communication with all levels of HCWs, and visibility.

Four nurse leaders (who also happened to be mothers) stood out as exhibiting the qualities of strong leadership that HCWs outlined. These four nurses expressed great concern for staff members (to the extent that it would keep them up at night) and worked hard to ensure all staff were protected, particularly staff from ethnic minorities, as this group was at increased risk. One nurse stated, "*I felt quite maternal, as in I wanted to protect everybody . . . So, trying to make sure that I was able to do the best that I could for people probably took over a little bit*". Another nurse increased staff sessions with a clinical psychologist and ensured all her staff attended. These leaders were aware of the impacts of COVID-19 on their staff's personal lives, such as those with young children who were home schooling and working at the same time. Notably, two of these nurses began planning for the pandemic in January, significantly before other leaders who participated in interviews.

Many HCWs were pleased with the dissolving of hierarchical and divisional boundaries and increased attention to the voices of lower-ranking HCWs during the COVID-19 pandemic. Some HCWs were happy with open communication from leaders at the top, while a few HCWs pointed out that lower-ranking HCWs were not given a voice in decisions. Additionally, not all areas were represented at the top and therefore involved in decision-making, which impacted HCWs' experiences on the ground. For example, one PT felt that there was not representation of musculoskeletal (MSK) physiotherapy, and therefore it was not recognized by decision-makers. Another consultant doctor felt that the voices of nurses were not heard at the start of the pandemic, and she attributed this to the fact that the field is predominantly female.

### 3.7. Beliefs and Values

HCWs seemed to source self-worth and identity from their roles, which led to psychological difficulties for some redeployed to lower ranks. For instance, one PT stated, "*I think for some people on the wards it was an extremely stressful time because there was such a change in their roles. Like you'd gone from being a senior member of staff in an MSK department basically to*

*a therapist assistant in an inpatient setting, so your whole value and your self-worth has changed a huge amount based on your change in roles. I know some people really, really struggled with that*". HCWs also emphasized the importance and satisfaction of providing a high level of care, and so the inability to perform as well and provide the same standard of care during COVID-19 (due to uncertainty around the disease, stretching of nurses, PPE, and limited contact with patients' families) had considerable repercussions for mental wellbeing and resilience.

HCWs further indicated their willingness to contribute to this crisis and understood working on the frontlines as their duty. For instance, one pharmacist stated, "*I personally felt okay about gowning up and doing what I needed to do because it's just part of the job,*" while others stated that this was "*just the way it goes*".

### 3.8. Gender and Sexism

Many HCWs were surprised when asked about gender. Sometimes, HCWs (both female and male) would state that they had never thought about gender impacting their experience before. Consistently, HCWs cited their role as the most important influence on experience: "*I definitely think it's far more to do with the profession than the gender*". One nurse mentioned the gendered distribution of roles in his facility, noting that nurses and junior staff tended to be female while leadership tended to be male.

The majority of participants stated that their gender did not influence their experiences as a healthcare worker during COVID-19. However, there were a number of exceptions. A few HCWs (male and female) noted that men are likely to have poorer outcomes from COVID-19, but this did not seem to be of serious concern among both white and ethnic minority males, described as "*a bit of a worry*". A few female HCWs noted that gender did not play a role in their experience as their field was female dominated (e.g., physical therapy), but that it might play a role in other areas, such as surgery and cardiology. A number of HCWs also stated that mothers are put in an especially difficult position in dealing with work and home demands.

Sexism was reported by a small number of HCWs. One female anaesthetist stated: "*We experience more sexism in COVID times than I've experienced in the last 5 years. It felt like going back to the early '90s in that female consultants who were with their male peers would not be introduced to the junior doctors as one of the ITU consultants for the day despite sitting amongst the other ITU consultants for the day. We'll be routinely overlooked for thanks by male intensive care consultant colleagues at the end of a busy shift. We'll be asked to do menial kind of junior doctor type tasks. There's a lot of, using a common parlance, mansplaining going on. I think many of us thought oh we're just being oversensitive but actually I think about a group of 12 of us ended up talking about it and it seemed to be a consistent theme that under threat culturally people reverted to the '90s and there was a huge amount of sexism*".

A GP also shared her experience of combined sexism and racism. This GP is mixed race—White British and Black African. She described her experience in which the clinical lead holding weekly meetings during COVID-19 would only discuss things on his own agenda without input from the staff in the meetings (the majority of whom were BAME). When she asked to discuss more clinical issues, he refused. She said, "*This old, white man deciding that his agenda should be the agenda of all these doctors and all these advanced nurse practitioners, the majority of whom are women*" and described the clinical lead as "*men who try to make me inferior*".

### 3.9. Caring Responsibilities

Having children was a considerable source of stress and anxiety for parents. Both mothers and fathers expressed anxiety about the possibility of bringing COVID-19 home to their families and described the extensive measures they took to keep their families safe (e.g., a routine of changing and showering when coming home, sleeping in a different bed). In expressing concerns about their children's wellbeing, mothers generally presented

a higher level of stress and anxiety compared to fathers. However, men in this sample expressed less emotion than women in general.

All HCWs experienced a change in routine due to the pandemic, but this change seemed more stressful for HCWs with children because they had to abruptly negotiate childcare. Those who were working part-time before the pandemic (all women with children) experienced a more "*exhausting*" transition as they took on more work hours. For example, one lead nurse went from working Monday through Thursday to working 80 h a week and had to rearrange childcare to accommodate this.

Some HCWs continued to send their small children to nursery, considering themselves "*lucky*". One HCW brought her small child to a nursery that remained open for only her child and one other. Other HCWs with children who did not have a stressful time managing childcare had either a husband or wife that was on sabbatical, maternity leave, or working from home and able to take care of the children. Some of those HCWs whose childcare remained available or whose partner could provide full-time care actually spent more time than usual with their children during COVID-19, due to changes in working patterns (e.g., working more night shifts) and lockdown measures that increased time with immediate family.

Many HCWs were not so "*lucky*" and found themselves unable to rely on usual forms of support because of lockdown restrictions. One HCW reported losing her normal childcare because of the stigma of being a HCW. Limits on support and high demands led to a constant juggling of responsibilities, planning of care week-by-week, and sleep deprivation (this was only reported by mothers).

While some HCWs believed that this burden of care would likely fall more on mothers, the reported experiences of HCWs did not reflect this. Care labour seemed to be divided practically, based on which partner was able to take care of the children, rather than by sex. For instance, among one couple, both took one day off a week to care for the children. A number of female HCWs had husbands working from home who took over most of the childcare, as their job flexibility allowed this. A few male HCWs' wives managed the children because they were either on sabbatical or maternity leave. There was only one exception where a male HCW's wife was simultaneously working and taking care of the children. While HCWs improvised and were creative in managing childcare, this was noted to be unsustainable, costly, draining, and not providing the preferred standard of care. HCWs' reports highlight the importance of partner support and financial and job security in allowing for a practical and more equitable division of household labour. Therefore, the experiences of childcare among this group of largely middle-class healthcare workers may not generalize to single parents or less privileged HCWs.

Lastly, some mothers reported feelings of disappointment and guilt about not being able to give enough care to their children because of work demands during COVID-19. None of the fathers in this sample expressed this. One nurse specialist shared her experience teaching her children, "*It's been difficult. I don't feel like I've been a good teacher in terms of I felt very unable to motivate them in the right way, to do their work in the ways that I would like them to do. It's all seemed a bit futile in places, so draining at home . . . and then quite draining coming to work . . . there's not that much reprieve*". Mothers also voiced feelings of guilt surrounding their inability to give their children sufficient attention and for relying upon the TV too much. One mother also expressed guilt for not being able to better support her family in general.

### 3.10. Pregnant Healthcare Workers

The sample included four healthcare workers (all doctors) who were either currently pregnant or recently pregnant during the first peak of the pandemic. The experience of one pregnant healthcare worker is illustrated in a vignette, found in Appendix C. One pregnant HCW worked at a private hospital, while the rest worked in the NHS. Experiences at the private hospital seemed substantially different, including a poorer maternity pay package (described as "*a bit of a hard one to swallow*").

All of these women were working at the start of the pandemic (before Public Health England announced that pregnant healthcare workers should be taken off the frontline). Two contracted COVID-19 and one was exposed early-on in her pregnancy while protected in PPE. Another, who worked at a private hospital, was exposed during her third trimester without any PPE, which made her feel, "*Really frustrated at the hospital . . . No one thought about putting any plans in or any PPE . . . I'm really cross but there's not a lot that I can do*".

These HCWs were supported in different ways throughout pregnancy. Two NHS HCWs were able to continue working safely due to the support of management and, in one case, the advocacy of a college tutor. Continuing to work had a positive impact on these HCWs' mental health. For instance, one HCW described her feelings about returning to work: "*I felt so much better, so much happier that I could actually contribute to this crisis that was going on [ . . . ] Now I'm allowed to do some clinical work again and . . . that's really helped. I feel a lot happier since that began, so I think I'm lucky in that I've had a very proactive department who supported me and found work for me to do. If I was stuck at home, it would be awful*".

One HCW who worked at a private hospital was not supported by her workplace, which amplified the anxiety and isolation that she felt. Another HCW did not publically disclose her pregnant status and continued working. She became ill with COVID-19 in the period leading up to the peak, but only took one day off of work and then continued working from home because: "*I was feeling a lot of pressure to support what was a very busy service . . . In that period, it felt very much like you couldn't let people down so you just kept going*". She then had a miscarriage after having COVID-19.

The process of attaining support at work led to a lack of privacy, input from colleagues, and a general sense of discomfort. For instance, one pregnant HCW tried to reduce attention to her pregnancy by "*not making a thing of it*" and suggested her co-workers appreciated this. Another HCW was frustrated by this lack of privacy and colleagues' discussions behind her back about whether or not she should be working. She was also moving to a new job soon and was concerned about showing up as "*the awkward employee with lots of complicated issues*".

Even when pregnant at home, these women retained their identity as doctors and the accompanying sense of duty, leaving them with guilt and disappointment. One HCW stated, "*When I first came off the rota, I just felt really guilty . . . Also, I've spent the last 9 years of my life training for this and it was like why suddenly am I stepping away from this and it felt like I should be there*". Another HCW shared her feelings of guilt during the weekly clap for carers, even though she was still working from home: "*I felt like I didn't deserve that applause because I wasn't on the frontline and I wasn't doing what I should be doing*".

Two HCWs gave birth during the pandemic. Social support was limited during this time due to the lockdown restrictions, which caused feelings of isolation. One HCW stated: "*The hardest part about all of it is it's been really quite lonely*" and, "*I've just been really down because you just want your mum*". These two HCWs were supported by their partners at home. In one case, this support was limited by short paternity leave, leading to an overwhelming burden of care.

## 4. Discussion

Examining HCWs' experiences through a gender lens highlights inequalities, the unique needs of working women, and policy implications that may otherwise be overlooked. The results above demonstrate a feminized burden of care during COVID-19, the need for meaningful support for nurses, conflicting cultural expectations of workers and mothers, gender blindness, and point towards necessary actions to support healthcare workers and strengthen the NHS.

### 4.1. A Feminized Burden of COVID-19 Care

While the majority of HCWs in the NHS are female, this solely does not account for the disproportionate burden of care shouldered by women in the NHS. Participants noted that the experiences of lower-ranking staff (who are more likely to be female) were relatively

more difficult than higher-ranking staff. Furthermore, the redeployment patterns discussed among this sample of HCWs occurred along pre-existing gendered divides. Given that HCWs in lower-ranking positions were more likely to be redeployed than those in higher-status positions, vertical segregation in the NHS may have widened during the pandemic. While present among surgical staff, physiotherapists, and the other healthcare workers in this sample, this pattern of redeployment may differ among other groups of HCWs, such as anaesthetists who were in high demand during the pandemic (Buck et al. 2020), and throughout different stages of the pandemic.

As supported by our results, pre-existing gender inequalities in the NHS laid the groundwork for redeployment patterns and differential experiences that result in a feminized burden of COVID-19 care. Women also account for the large majority of nurses, the group identified as having the most stressful and difficult time during this pandemic. These pre-existing conditions resulted in redeployment patterns and differential experiences that put women at higher risk for exposure to COVID-19 and mental health issues arising from the emotional and traumatic experiences on the frontlines. This risk is especially high among ethnic minority women, as ethnic minority groups are disproportionately located in lower-ranking positions in the NHS (Chaudry et al. 2020), and a significant proportion of NHS nurses identify as BAME.

Global research findings that female HCWs had a higher prevalence of posttraumatic stress symptoms should therefore consider the increased pressures on female HCWs, rather than simply attributing this to the normally higher rates of these symptoms amongst females (Pappa et al. 2020). In parallel to increased mental health support, protection, material support, and gender equality in the workplace are crucial in maintaining the mental wellbeing of HCWs.

### 4.2. Risk

The groups mentioned by participants as having inadequate protection were AHPs, healthcare assistants, and cleaning staff—all dominated by women (NHS Digital 2018b; HCPC 2020). Participants also mentioned nurses (in general, not specifically at their facility) as a group without adequate access to PPE. A survey of over 3000 UK nurses found that 67% of participants did not have adequate PPE during the COVID-19 pandemic (Dean 2020). BAME nurses were found to have less access to adequate PPE than their white counterparts (Royal College of Nursing  RCN). Insufficient access to PPE among nurses is concerning given their close and prolonged contact with COVID-19 patients and the high proportion of BAME nurses (given the increased risk of severe COVID-19 outcomes among BAME groups).

Importantly, PPE shaped for the male body may also increase the risk of COVID-19 exposure among females (Hoernke et al. 2020). The issue of PPE not fitting women has been raised among female NHS staff throughout the pandemic (Porterfield 2020; Pugh 2020) and a recent study showed that, compared to males, females are less likely to fit FFP3 masks (Ascott et al. 2020). Male bias in PPE was an established problem prior to the pandemic (TUC 2017) and remains to be addressed.

### 4.3. The Stretching of Nurses

NHS nurses were over-burdened, stretched, and under-represented in decision-making during the COVID-19 pandemic. These results resonate with findings from the severe acute respiratory syndrome (SARS) outbreak, in which nurses in Taiwan were at heightened risk due to close proximity to patients and long work hours (Shih et al. 2007). Taiwanese nurses also suffered significant psychological challenges and were not sufficiently protected, leading nurses to feel as if they were being sacrificed (Shih et al. 2007). In China, nurses were identified as having the highest stress levels compared to doctors and healthcare assistants, stemming from feelings of a lack of control, vulnerability, and fears of contracting and spreading the virus (Wong et al. 2005).

The experiences of NHS nurses during the COVID-19 pandemic can be understood within the gendered history of healthcare roles and hierarchies. Professional nursing emerged in Britain as a feminine role, defined by traditional, socially ascribed feminine traits and both subordinate and complementary to doctors (Gamarnikow 1990), while medicine and the doctor emerged as masculine (Davies 2003). Gender bias continues to influence the differential description and valuation of medicine and nursing (George 2007, 2008) and therefore indirectly impacts the personal experiences of HCWs. There has been a chronic shortage of nurses in the NHS for decades, which is partly attributed to a lack of sufficient investment in nursing higher education, low compensation, staff feeling that they are not valued, and poor retention (Truswell 2020; Finlayson et al. 2002). In order to be prepared for future health emergencies, it is imperative to increase funding and support for nurses and nurse education. Globally, there is a need for an increased valuation of nursing as well as recognition and rectification of the gender biases that continue to devalue care work.

### 4.4. The Ideal Worker

A gender lens framework illuminates the structural roots of individual problems; as stated by Gerson (2004), "Although work-family conflicts are experienced in intensely personal ways, they have institutional sources" (p. 165). Feelings of guilt and disappointment among working mothers and pregnant HCWs can be understood as resulting from the conflicting models of worker and mother in Western society. The contemporary model of an ideal worker expects that workers give the bulk of their time and energy to their work, valorising full-time and overtime work (Greenberg et al. 2009). This model conflicts with the predominant model of motherhood in Western countries, which expects mothers to invest high levels of emotional labour, time, and energy into raising their children (Scarborough and Risman 2017). The maternal body thus emerges as a site of contestation between dominant maternal and worker ideals.

Pregnant HCWs and working mothers were placed in difficult positions by the gendered expectations of being a 'good healthcare worker' that were amplified during this crisis. While being a 'good mother' implies prioritizing one's baby or children, work pressures amplified in this crisis ask the mother to put her body and children at risk for the benefit of patients. This tension is seen among pregnant HCWs in their consistent emphasis on wanting to work and contribute, as well as retainment of their identity as doctors and the accompanying sense of duty while at home.

Pregnant HCWs had to sacrifice work and career development for their babies; the one exception sacrificed her maternal wellbeing and time with her family for work, losing her baby in the process. Working mothers had to sacrifice ideal care for their children for work. While fathers also made sacrifices, they did not express the same emotional impact. These findings thus reflect gender differences in socially prescribed parental and worker expectations that may affect individual emotions.

Guilt implies an unwanted event with the self as the cause (Hochschild 2003). Rather than assigning blame to unequal gendered expectations of mothers and workers or the COVID-19 pandemic, mothers held themselves responsible for not fulfilling these competing demands. This calls for a new conceptualization of motherhood in the workplace that understands the psychological conflicts that working mothers face not as due to their individual choices, but rather norms that are biased towards men and thus discriminate against women (Williams 2000).

### 4.5. Pregnant Healthcare Workers and the Othering of the Maternal Body

Our findings also point to the continuing discrimination against pregnant women in the workplace in England (Greenberg et al. 2009). Reports from pregnant HCWs imply an othering of the maternal body and its discordant position within healthcare structures and norms, specifically among doctors. The othering of the maternal body elucidates the contradictions between work and motherhood, as the female body itself, particularly when

pregnant, is seen as "other" and "disruptive" in professional environments characterized by masculinized worker norms such as "bodily control and stability" (Hennekam et al. 2019, p. 917; Greenberg et al. 2009). In this case, the masculinized domain of medicine and especially the expectations of doctors in a health crisis exacerbated the conflict and ensuing guilt of pregnant doctors.

In addition, the exposures of multiple pregnant HCWs at the beginning of COVID-19 call for immediate attention to the needs of pregnant HCWs in the workplace, especially within private hospitals. Pregnant HCWs were only able to work from home safely if they were supported by their organization to do so. The ability to continue working from home seemed to substantially improve pregnant HCWs' mental health.

### 4.6. Gender Blindness

Although gender impacts how healthcare is understood and valued as well as the position and experiences of individuals, gender biases can be concealed by description biases that impact how roles and healthcare work are understood (George 2007). While our findings show that gender impacts individual experiences, the general consensus among participants was that it does not. This belief also seemed to be prominent among top healthcare officials, as no policy in the health and social care system response to manage COVID-19 addressed gender differences or inequalities (The Health Foundation 2020).

Other research has also found a lack of gender awareness among medical staff, which can be attributed to factors including the unconscious nature of gender stereotyping (Davies 2003), the emphasis on objectivity in medicine and resulting tendency to understand oneself as gender-neutral, and a lack of understanding of the social construction of gender (Risberg 2004). Gender insensitivity has been found to accompany discrimination and gender bias among medical professionals and can lead to the reproduction of gender inequalities and norms (Risberg 2004; Risberg et al. 2006). Training on gender in relation to HCWs, organizations, and patients should therefore be integrated into the education of all health professionals.

### 4.7. Gender Matters

For most HCWs interviewed, gender operated in largely silent, structural, and symbolic ways rather than explicitly in personal interactions. Although they claimed that gender did not matter, HCWs also asserted that the role defined experiences during the COVID-19 pandemic, yet these roles are themselves gendered. Gender has been replaced by job title, but the same gendered expectations are reinscribed in job descriptions and influence the valuation, support, and power given to different roles. Male bias in medicine continues, as seen in PPE shaped for the male body, experiences of sexism, and organizational norms that disadvantage working mothers.

As explicitly narrated by the majority of participants, gender did not independently affect experience within roles but rather indirectly through its influence on the nature of the roles themselves, placement within healthcare structure, and organizational culture. Conceptualizing gender as a social structure (Risman 2004) directs attention to how individuals discursively construct or resist hegemonic gender ideals within social contexts. HCWs on the ground interact and co-construct identities in ways that were perceived as equitable, producing gender as seemingly unimportant in shaping experience. This is seen in HCWs' responses that gender did not impact their experience, the dissolving of hierarchical boundaries, and the gender-neutral model of home labour. However, at the structural level, gender persists as a significant yet silent social stratifier that shapes access to resources and power.

### 4.8. Supporting our NHS Heroes

Our findings show that support for healthcare workers must extend beyond the individual worker to encompass family and caring responsibilities. Greater support for HCWs from the English government and family-friendly policies in healthcare institutions

(e.g., an increase in paternity pay) are especially vital during COVID-19, due to the lock-down restrictions that limit social support. Policies that support an equitable distribution of childcare and home labour are necessary to support the efforts made by these HCWs towards a gender-neutral distribution of home labour.

Additionally, mental health support must be made available and accessible to all healthcare workers, especially redeployed HCWs and nurses, as well as those working or isolating at home, such as pregnant HCWs. Safety and adequate, properly-fitted protection must also be considered as foundational in supporting HCWs' mental health and wellbeing.

Lastly, increasing female leadership, especially female ethnic minority leadership, in the NHS is a simple and crucial step towards gender equality. Our results highlight the strong leadership exhibited by nurse leaders, adding to literature worldwide documenting strong female leadership during the COVID-19 pandemic (Coscieme et al. 2020; Sergent and Stajkovic 2020).

### 4.9. Strengths and Limitations

The most prominent strength of this research is its qualitative approach. As most research on gender among HCWs takes a quantitative or top-down approach (focusing on larger systems and structural inequalities), highlighting the voices and analysing the personal experiences of HCWs adds a valuable contribution to this emerging field.

There are also important limitations to this study. Firstly, the sample disproportionately consists of higher-ranking and White British HCWs, which restricts the experiences we had access to and therefore limits the applicability and generalizability of our findings. Our ability to take an intersectional stance in this gender analysis was also therefore limited. Secondly, our sample mainly consists of individuals from three trusts in England and the sampling was neither random nor representative. These findings, therefore, may not generalize to the experiences of other HCWs in England or the UK. Thirdly, all participants were either female or male and heterosexual, leaving out the experiences of other sexual orientations and genders. Fourthly, the low number of men in our sample limited our ability to examine the unique experiences of men and engage in meaningful comparisons between men and women. Lastly, gender is a highly complex concept that is difficult to investigate through telephone interviews with only a handful of questions. Therefore, this study does not provide a comprehensive account of the influence of gender on HCWs' experiences.

### 5. Conclusions

Our work demonstrates the importance of addressing gender when examining professional experiences and epidemics. In employing a gender lens framework, we expose how larger structural inequalities shape individual experiences and problems in meaningful yet often inconspicuous ways.

Pre-existing research has established gender inequalities and forms of gender segregation within healthcare systems. Our findings expose the dangers of these pre-existing inequalities by exhibiting how they are exacerbated during a crisis, contributing to a disproportionate burden of care shouldered by women. Similarly, our results add to feminist and social science research on the division of labour, the ideal worker, and the maternal body by illustrating how the pressures that working women face were intensified during the COVID-19 pandemic. These findings widen the scope of conversations around support of HCWs during pandemics to include family-friendly policies, attention to the needs of pregnant HCWs, and the significance of HCWs' social contexts. Results also point to both immediate steps and long-term cultural shifts necessary to improve gender equality, the wellbeing of healthcare workers, and therefore the strength of the NHS.

Our analysis shows that gender is significant when understanding HCWs' experiences, whether in a pandemic or not (see Figure 1). This is noteworthy in the context of a lack of attention to gender both among individual HCWs and in research and policy concerning HCWs in the UK. Although researchers are increasingly considering gender in health

research, this research often focuses on the patients rather than the workers. How can we expect to have equal health outcomes if the structure of our health system itself is unequal?

**National Policy**

- lack of support for HCWs with children that increased worry and stress among parents
- inadquate paternity leave placed stress on new mothers and fathers
- no pre-existing emergency safety guidelines for pregnant HCWs; measures put in place too late
- poor representation of women in health policy and decision-making
- no health and social care policies addressed gender inequalities or differences among HCWs

**Organizational Structure**

- feminized burden of care during COVID-19
- women form 89% of NHS nurses, the group with the greatest difficulties during COVID-19
- ethnic minority women are doubly marginalized in the structure of the NHS
- women under-represented in senior positions in the NHS, especially ethnic minority women

**GENDER MATTERS**

**Culture and Discourse**

- conflicting ideal worker norms and traditional cultural expectations of mothers place working mothers in a difficult position, leading to guilt and anxiety, especially among pregnant HCWs
- the pregnant body is discordant with medical organizational norms
- PPE designed for the male body

**Individual Experiences**

- feelings of guilt among mothers and pregnant HCWs
- higher burden of stress and mental health risks among redeployed, lower-ranking, and nursing staff (all more likely to be women)
- sexism is an issue in certain medical specialities
- ill-fitting PPE puts women at potentially higher risk of exposure

**Figure 1.** Gender Matters: summary of findings.

A crisis provides the valuable opportunity to reconstruct more equitable systems and nations. As we continue to respond to the COVID-19 pandemic and develop strategies to support staff, we should engage in meaningful conversations on the role of gender in the daily practices of HCWs. Future research on HCWs' experiences should, when possible, take gender and other social stratifiers into account. If not, we risk reproducing existing inequalities and invisibilising the specific needs of female HCWs.

**Author Contributions:** Conceptualization of the study C.V.-P., conceptualization of analysis N.R. and C.V.-P., methodology C.V.-P. and N.R., data analysis N.R. and C.V.-P., writing N.R. and C.V.-P. Both authors have read and agreed to the published version of the manuscript.

**Funding:** This research received no external funding.

**Institutional Review Board Statement:** The study was reviewed and approved by the Health Research Authority (IRAS: 282069) and the R&D offices of the hospitals where the study took place.

**Informed Consent Statement:** Informed consent was obtained from all subjects involved in the study.

**Data Availability Statement:** Data are contained within the article.

**Acknowledgments:** We would like to thank Joseph Calabrese, for his supervision and review of the MSc dissertation from which this analysis is based, and Anna Dowrick, for her input and discussion

around pregnancy and emotions. We would also like to thank Anna Badley, Caroline Buck, Kirsi Sumray, Georgina Singleton, Aron Syverson, Lucy Mitchinson, Harrison Filmore, Louisa Manby, Anna Dowrick, and Sasha Lewis-Jackson for their support in data collection. We are also grateful for the healthcare workers who shared their experiences in interviews.

**Conflicts of Interest:** The authors declare no conflict of interest.

## Appendix A. Interview Guide

First, I want to ask you about your work and the services you provide.

1. Background: Can you tell me about your role?
   - Can you tell me a bit about your role? (e.g., Daily tasks, department, responsibilities)

2. Have you been in contact with patients who had suspected and/or confirmed COVID-19? Probes:
   - In what capacity?
   - How have you found working around these patients?
   - PPE physical effects? (e.g., dehydration, discomfort, restriction in movement, difficulties communicating)
   - How has PPE impacted the type of care you provide patients?
   - What psychological/emotional impact did this have on you?

3. How has the COVID-19 outbreak affected health services in your department? Probes:
   - How has this affected your normal daily tasks/responsibilities? Change of role?
   - Impact of COVID-19 on the delivery of services to non-COVID-19+ patients (i.e., cancellation of elective surgeries)
   - What tasks are you able to do more or less effectively?
   - How do you manage the isolation of suspected cases and confirmed cases?
   - Has there been appropriate transfer of patients within and out of hospital?
   - Has there been an impact on staff's ability to make diagnoses and act on them?
   - Has the supply of drugs, equipment and PPE been affected?
   - Have staff been redeployed from or within your health facility

4. What were the preparedness strategies implemented locally (department, hospital or Trust)?
   - Did you feel these strategies were enough?
   - What do you feel was particularly successful?
   - Should the Trust have prepared differently?
   - Did you receive any training? (including but not limited to PPE training such as mental health and well-being training)
   - Did you have access to guidance on PPE?

5. Do you currently have any concerns or fears in relation to . . .
   - Work (response efforts, PPE, services)
   - The national effort

6. Over the past months, have you experienced any problems with aspects of your daily life such as sleeping, eating, concentration, or additional worries or anxiety?

7. Mental health support (to address risk of moral injury, trauma and developing severe mental health problems)
   - Are you aware of any support available for staff wellbeing and mental health?
   - Have you had the opportunity to talk about your mental health with your supervisor/team leader?
   - Have you had worrying experiences in the last week? Did you receive support after? If so, what type of support? (including formal and informal support)
   - Interactions between peers: Do you have time to socialise with your team? What has changed with COVID-19?

8. Have you been involved in caring for patients who are dying or expected to die soon?

a.   (If relevant based on previous discussion) Can you please tell me about the palliative care tasks you are involved in with COVID-19 patient?

Ask about each of these specifically:

- Advanced care planning
- Symptom management and patient comfort at end of life.
- End of life decision making (e.g., triage of limited equipment)
- Working with families (e.g., updating on health, organising visits)

  ○   How have you found these tasks? (e.g., difficulties? patients' reactions? preparedness? what works well?)
  ○   Was this part of your normal role prior to COVID-19?
  ○   What difficulties have you faced in these tasks?
  ○   How does this differ to normal palliative care?
  ○   How much choice do patients have?
  ○   What are the rules/policies relating to this? Do you feel these are suitable?
  ○   Was there training or support available relating to this?
  ○   Do you feel this has had an emotional impact on you?

9.  What do you feel is most important to offer COVID-19 patients at end of life and their families?

  ○   What is working well?
  ○   What should we do more of?
  ○   What can we improve?
  ○   What support do we need to offer HCW delivering palliative care?
  ○   Do you have any concerns for the future?
  ○   Are you able to offer bereavement support to families?

10. OTs/PTs and others in charge of rehab: What are your main concerns about the impact of COVID-19 on the body (e.g., muscle degeneration, dexterity, impact to the lungs etc.)?

- What resources do you have to deliver rehabilitation care? - ask their opinions on the Mary Seacole rehab hospital
- Is there a difference in resources for COVID-19 and non-COVID-19 patients?
- redeployment:
- did you feel prepared to deliver respiratory care?

11. (If relevant based on previous discussion) Can you please tell me about the rehabilitation care tasks you are involved in with recovered COVID-19 patients?

- Have you received any guidance on how to deliver rehabilitation services to recovered COVID-19 patients?
- do you feel prepared to deliver this care?
- do you have any concerns about capacity to deliver this care?
- OT: How does this differ from normal rehabilitation care, e.g., delivering care at home?
- OT: How has COVID-19 impacted your contact with patients?
- Has the pandemic impacted the flow of your patients through hospital e.g., are more or less patients being discharged to homes and bed-based rehab?—What is the impact of this?
- How do you think your role will be impacted as a growing number of people will need rehabilitation? Any concerns?

Discharge:

- What criteria do you use to decide to discharge a covid patient?
- Where do patients usually get discharged (e.g., home or rehabilitation centres?
- How do you communicate to patients and their families the care they will need?

12. How have health services been strengthened, or how could they be strengthened during the outbreak? Probes:

- Support to HCWs from the health system and partners?
- Capacity for rapid response
- Policies? e.g., Guidance and emergency protocols?
- What would help HCWs to maintain normal services as well as COVID related services?
- If GP: Health promotion and community engagement. How?
- If GP: Linkage to other support organisations, e.g., charities, schools?

13. Is there anything you feel should be changed to make health services more effective in future emergencies? Probes:

- Support to HCWs? From whom and How?
- Coordination and official guidance of COVID-19 response.
- Early detection and reporting.
- On-going health promotion and community education, e.g., potential sources of infection, safe practice?
- Mobilisation? e.g., identifying and coordinating trusted community volunteers and support?
- Disease outbreak control activities?
- Testing (public and staff)

14. Do you feel your experience has been different from other HCWs? Does gender play a role? How about ethnicity?

15. How has your life at home been impacted by COVID-19?

16. Do you have any caring responsibilities, such as children or elderly family members? If yes:

    a. How are you managing care during the COVID-19 pandemic?
    b. (If they have children) How has being a HCW during the pandemic impacted your ability to parent?
    c. What fears, worries, or emotions arise from the responsibility of caring for others during this time?

17. Are you pregnant?

    a. If so, how has this impacted your work and experience as a HCW during the COVID-19 pandemic?

18. Is there anything else you would like to mention that you feel is important?

    Thank you for your time and for sharing your opinions and experiences with us.

**Appendix B. Coding Framework**

**Node (sub-node)**

- Access to resources (training and preparation)
- Beliefs and values
- Burden on nurses
- Caring responsibilities
- Compensation
- Coping strategies
- Division of labour at home
- Division of labour at work (care work and emotional labour)
- Emotions (guilt; worries, concerns, anxieties)
- Gender differences (lack of an impact of gender)
- Impact of ethnicity
- Impact on career development
- Impact on home life
- Leadership and decision-making
- Media and social media
- Protection of staff

- Sacrifice
- Social norms and ideologies
- Social support
- Team and co-worker dynamics
- Work/life balance (changes to working patterns)

**Appendix C. Vignette of a Pregnant Healthcare Worker**

This pregnant healthcare worker is in her late 30s, British Indian, a Locum Consultant at an NHS trust in London, is married to a husband with a stable income, and has a 3-year-old (whose nursery closed). At the beginning of the pandemic, she was still working and was exposed to two patients that ended up testing positive for COVID-19, but she felt protected in PPE. She continued working, "*even though there was a small part of me that was slightly niggling—not so much the risk to myself . . . but just that potential what if I did something to my unborn baby and the rest of my family*". Even with that concern, she kept working, thinking "*I'm not going to make a fuss, I'm just going to get on with it*".

She was reluctant to leave work and, in conversation with her manager, decided to work from home a day after Public Health England (PHE) announced that pregnant women should shield. Her colleagues were very supportive, both when she was coming to work and then shielding. She states, "*I think they probably recognised that I was being quite—I don't know if brave is the right word—but I was kind of not making a thing of it and for as long as possible I was just coming in and planning to come in and carrying on doing my job*". She was able to work from home, but still felt guilty about not being on the frontlines.

She had a difficult time once she had the baby, especially once her husband's paternity leave was over: "It was too hard like my new baby was waking up every hour-and-a-half and then you'd wake up so I'd basically get maybe 3 or 4 broken hours of sleep a night. Feeding all day, running around after the 3-year-old who obviously had loads of energy . . . I think we were really struggling, and it was fine while my husband was technically off on paternity, so we coped for the first couple of weeks, but then he had to go back to work full-time and it was just impossible". She recognized that this stress was not only impacting her, but also her husband. She made a difficult decision and went to stay with her in-laws (against COVID-19 lockdown rules) where she had help and support.

For her, the most upsetting part of her experience was leaving work. She describes this, "To be honest I was really disappointed. I really wanted to be in the thick of it. It felt like everything I've been training for for the last 13 or 14 or however many years it's been . . . it was a really exciting time to be in respiratory medicine, a new disease, to be on the cutting edge of it, to be learning about this disease and then to have that taken away, I found it really upsetting". She felt as if this experience was important to her career and tried her best to keep up with the virus through online webinars, but still feared she would be behind when returning to work.

Her selflessness really stood out throughout the interview. Even though she acknowledged that she was at higher risk because of her ethnicity, her concern consistently was with her patients, family, and baby's safety and not with herself.

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
