# Peer review of "Gender Matters: A Gender Analysis of Healthcare Workers’ Experiences during the First COVID-19 Pandemic Peak in England"

_socsci, doi:10.3390/socsci10020043_

Round 1
Reviewer 1 Report
Thank you for the opportunity to read this timely paper about the impact of the recent pandemic on HCWs in England. I appreciate the contribution this manuscript makes to our understanding of how the current health situation tries the English health care system and its frontline workers. The paper is clearly written and generally persuasive, although the authors' claim to be using a 'gender lens' is not accurate. To do so is to present a phenomenological review of how gender shapes the lived experiences and understandings that people have of their day to day activities. The article, as presented, is a structural review of the pandemic and how it differentially impacts men and women in health care. To say that more women are nurses and in the frontlines, and more women are affected by disruptions to child care, is to provide comment on structural inequalities related to gender. Indeed, the participants in this study did not seem to view their experiences as being heavily gendered. The authors state that "many HCWs were surprised when we asked about gender" and that "the majority of participants stated that their gender did not influence their experiences as a healthcare worker during COVID-19." As well, the authors do not make any sort of direct comparison between the lived experiences and understandings of men and women in this sample. The authors are more accurate in claiming that their paper is about a "feminized burden of COVID-19 care" and use the interviews to detail this structural inequality that is gendered. Thus, the paper should be rewritten to align what is done and its framing. Key claims need to be modified. I would also suggest removing language that is inherently statistical - for qualitative work, for instance, use of "significant" is not appropriate.
Good luck to the authors in their
Reviewer 2 Report
Further precision suggested for study description, results
